# Assessment of cognitive functioning after living kidney donation: A cross-sectional pilot study

**Marie Mikuteit**[1☺], **Faikah Gueler**[2☺], **Iris Pollmann**[3¤], **Henning Pflugrad**[1], **Meike Dirks**[1], **Martina de Zwaan**[3‡*], **Karin Weissenborn**[1‡]

1 Department of Neurology, Hannover Medical School, Hannover, Germany, 2 Department of Nephrology and Hypertension, Hannover Medical School, Hannover, Germany, 3 Department of Psychosomatic Medicine and Psychotherapy, Hannover Medical School, Hannover, Germany

☺ These authors contributed equally to this work.
¤ Current address: Department of Psychosomatics and Psychotherapy, University Hospital Schleswig-Holstein, Kiel, Germany
‡ MZ and KW also contributed equally to this work.
* dezwaan.martina@mh-hannover.de

**Data Availability Statement:** All relevant data are within the manuscript and its Supporting Information files.

## Abstract

### Background

Chronic kidney disease (CKD) has emerged as a risk factor for cognitive impairment. Living kidney donation results in reduction of the donors' renal function. This is considered acceptable in general but possible associations with cognitive function have not yet been studied.

### Methods

Sixty living kidney donors (LKD), who had donated between 2003 and 2012 at Hannover Medical School, underwent neurocognitive testing including attentional and memory testing. In a cross-sectional design results were compared with data of healthy controls (n = 40) and with norm data given in the respective test manuals adjusted for age, sex, and education.

### Results

The median age of the LKD was 58 (range 39–70) years and the median time since donation was 7 (range 4–14) years. The LKD did not differ from controls in most of the cognitive test results and a composite attention test sum score. However, LKD did worse than controls in tests of working memory, parallel processing of stimuli, and sustained attention. No differences were found regarding quality of life. In LKD cognitive test results correlated significantly only with educational level but not with time since transplantation, eGFR, somatic comorbidity, quality of life and levels of fatigue, distress, depression, and anxiety.

### Conclusions

Our data show a fairly normal performance of LKD in most attentional and memory tests. However, our pilot study also suggests some cognitive impairment in attention tests in LKD which would need to be confirmed in longitudinal prospective studies.

**Funding:** MM received a stipend from the doctoral program "KlinStrucMed" at Hannover Medical school which is funded by the Else-Kröner-Fresenius Stiftung (https://www.ekfs.de/wissenschaftliche-foerderung/aktuelle-foerderungen/klinstrucmed). The funders had no role in study design, data collection and analysis, decision to publish, or preparation of the manuscript.

**Competing interests:** The authors have declared that no competing interests exist.

## Introduction

Kidney transplantation is the preferential treatment for patients with end stage renal disease (ESRD) [1]. The outcome for the recipients is in general better with a living donation than with deceased donation [2]. Living kidney donors (LKD) rarely experience severe side effects of the nephrectomy such as ESRD or complications like-re-hospitalization [3, 4]. The mortality of LKD does not differ from the general population [1, 5]. In most cases the donors' renal function remains stable for a long time [2, 5–7] though at a decreased level compared to before donation. In the long term, between 12 and 25 percent of the LKD develop an estimated glomerular filtration rate (eGFR) of < 60 ml/min [2, 8].

A systematic review revealed that most, but not all, cross-sectional and longitudinal studies suggest an association between cognitive impairment and CKD [9]. Even though cognitive impairment is most likely to occur at eGFRs $< 30$ ml/min/1.73 m$^2$ [10], it has been demonstrated in community-based studies that cognitive functioning is reduced even in subjects with only moderate CKD, e.g. with an eGFR between 30 and 60 ml/min/1.73 m$^2$ [11, 12]. Cardiovascular and other risk factors have shown to mediate the relationship between CKD in some but not all cognitive tests [12, 13]. Even after controlling for traditional cardiovascular risk factors, patients with CKD showed worse global cognition, visual-spatial orientation, concentration and memory compared to controls suggesting that CKD might be an independent risk factor for cognitive decline [12]. To the best of our knowledge, there are no available data on cognitive functioning in LKD even though their eGFR frequently declines post-donation.

Thus, the goal of the present cross-sectional pilot study was to assess cognitive functioning in LKD and to compare the results in LKD with those of heathy controls and with norm data presented in the respective test manuals. We hypothesized that LKD would show worse cognitive functioning compared to healthy controls. Our secondary hypotheses were that impairment of cognitive function in LKD would be associated with lower eGFR, higher levels of fatigue, depression, anxiety, and general psychological distress and lower levels of quality of life.

## Material and methods

### Study population

We investigated a convenience sample of 60 LKD out of a total sample of 315 LKD who participated in a follow-up study assessing physical and mental well-being after living kidney donation. Participants had undergone donor nephrectomy at Hannover Medical School between 2003 and 2012 [14, 15]. Exclusion criteria for this neurocognitive sub-study were any neurological disease and any mental disorder, use of central nervous system (CNS) affecting medication, language barrier, and age > 70 years.

Forty healthy subjects served as controls. They were recruited in the environment of participating physicians and patients and as a group did not differ with regard to sex distribution, age, and educational level from the LKD sample.

The study complied with the Declaration of Helsinki and was approved by the ethics committee of Hannover Medical School (no. 3252–2016). All patients gave their written informed consent.

### Neurocognitive testing

The Test of Attentional Performance (TAP) was used [16, 17]. It is comprised of a collection of methods which allows a differentiated diagnosis of attention deficits. The TAP is a standardized software package that uses simple reaction paradigms in which one has to react to well discriminable, non-verbal stimuli by a simple key press. The performance criteria are the

reaction time (RT) in milliseconds (ms) and any mistakes. The present study utilized the following eight subtests: (1) the alertness test (assesses the increase in level of attention when anticipating a stimulus), (2) the working memory task (probes the ability to control information flow and update information in working memory in real time), (3) the crossmodal integration task (examines the ability to detect the pre-specified combination of an acoustic stimulus and a subsequent visual stimulus), (4) the flexible reaction test (a set-shifting task that requires alternating reactions to numbers and letters), (5) the divided attention test (assesses ability to process visual and auditory stimuli in parallel), (6) the Go/No Go task (assesses the ability to suppress an inadequate response and therefore executive attention), (7) the incompatibility test (assesses the effect of contradictory stimulus information on stimulus processing), and (8) the covert shift of attention task (assesses the ability to focus visual attention) [16, 17]. Additionally, the cancelling d test [18] for the assessment of sustained attention was used. Finally, the following tests were utilized to assess memory functioning: the Recurring Figures Test (RFT) [19] and the Word Figure Memory Test (WFMT) [20]. The RFT assesses learning ability and recognition of nonverbal material and the WFMT assesses recognition of words and figures separately.

For the cancelling d test, the TAP battery tests and the RFT, results equal to or below the 10th percentile of norm data were considered abnormal, for the WFMT a z-score $\leq$ -1.3 compared to norm data was considered as abnormal. The results of the subtests of the TAP battery and the cancelling d test were used for the calculation of a composite attention test sum score, which gives the rate of abnormal test results out of the total number of attention test results achieved (range 0 to 1) and depicts a representative score for each patient's individual attention ability [21–23]. An attention test sum score > 0.4 was considered to represent a clinically relevant cognitive impairment. If a participant was not able to complete a subtest, this was counted as abnormal result. The reasons were lack of comprehension, a very long reaction time and a large number of mistakes. Other putative reasons such as decreased visual acuity were excluded.

To control for unexpected study-inherent confounding factors such as incorrect test instructions, for example, the patients' test results were compared to the results of 40 concomitantly examined healthy controls in addition to the comparison with pre-defined norm data.

All cognitive assessments were performed by members of the working group according to a predetermined procedure. The cognitive tests were conducted in an undisturbed environment either at the hospital (LKD) or at other places such as private homes or work places (controls). The tests took around 2 hours per participant and were always completed in the same order.

## Questionnaires

To assess quality of life the Short-Form 12 Health Survey (SF-12), a short version of the SF-36 Health Survey, was used and adjusted to American standards [24, 25] in participating LKD and in controls.

In LKD the presence and extent of fatigue was assessed using the Multidimensional Fatigue Inventory (MFI-2) [26, 27]. Symptoms of depression were assessed with the 9-item Patient Health Questionnaire-Depression Scale (PHQ-9) [28] and symptoms of anxiety were assessed with the 7-item Generalized Anxiety Scale (GAD-7) [29, 30]. The one-dimensional short version of the Symptom Checklist 90 (SCL-9) [31] was applied for assessing general psychological distress (global severity index, GSI).

## Sociodemographic and clinical data

The survey also contained investigator-generated questions related to personal data of the donor and donation-specific variables such as age, sex, educational level, somatic

comorbidities and date of the donation. Additionally, kidney function (eGFR CKD-EPI) of the 60 donors was assessed at the time of the study. Two LKD refused blood sampling at the time of study assessment.

## Statistical analyses

All statistical analyses were performed using IBM SPSS Statistics, version 24. Depending on data distribution, either the Student's t test or the Mann-Whitney U test was used to conduct between-group comparisons (LKD versus controls). Chi-square tests were conducted for categorical data. Effect sizes were calculated with Cohen's *d and phi* [32]. Regarding Cohen's d, 0.2 expresses a small effect, 0.5 a medium, and 0.8 a large effect. Regarding phi (df = 1), 0.1 expresses a small effect, 0.3 a medium, and 0.5 a large effect. Medians and the 25th and 75th percentiles and number of participants and the percentages of participants are given for individual variables.

In the LKD group we performed regression analyses to examine putative associations between the attention test sum score and sociodemographic and clinical variables. Univariate linear regression analyses were performed with the attention test sum score as the dependent variable and age, sex, educational level, time since donation, current eGFR, change in eGFR from pre-to post donation, the scores of the MFI, PCS, MCS, PHQ-9, GAD-7, and SCL-9, and the presence of hyperlipidemia, coronary heart disease, diabetes, hypertension, and hypothyroidism as independent variables.

A p-value <0.05 was regarded as significant. Since the study was primarily exploratory, we did not correct for multiple testing.

## Results

### Baseline characteristics of donors and controls

The comparison between LKD and controls are summarized in Table 1. Thirty-four (56.7%) of the 60 donors who participated in our study were female. Median age at the time of assessment was 58 years in the LKD group and 55.5 years in the control group. The median time since donation was 7 (range 4–14) years. Donors had attended school for 11 (range 8–13) years in median, controls for 12 (range 9–13) years. Regarding sex, age and years of education there were no significant differences between LKD and healthy controls. Significantly more LKD than controls were diagnosed with hypertension and hyperlipidemia. The frequency of hypothyroidism, coronary heart disease and diabetes mellitus did not differ between groups. Additionally, LKD and controls showed scores within the normal range for quality of life.

### Comparison between participants and non-participants

Of the 315 LKD who participated in the follow-up study, 184 LKD met inclusion criteria for our neurocognitive pilot study. The 60 LKD who participated in the neurocognitive study did not differ with regard to age, sex, time since donation, and the scores of the assessment instruments for health-related quality of life, mood, and fatigue from the non-participants (data not shown). The 60 participating LKD reported a higher educational level compared to the non-participants: median 11 years (25th; 75th percentile 10.0; 12.8) versus median 10 years (25th; 75th percentile 9.0; 12.0) (p = .034).

### Attention and memory testing

LKD and controls did not differ regarding the attention test sum score (Table 2). There were also no differences between the two groups in most of the subtests. However, donors showed

**Table 1. Comparison between living kidney donors (LKD) and controls.**

|  | LKD N = 60 | Controls N = 40 | p-value |
|---|---|---|---|
| Sex (female); N (%) | 34 (56.7) | 20 (50.0) | .512 |
| Age at time of assessment (yrs); median (range) | 58 (39–70) | 55.5 (35–70) | .089[1] |
| Age at time of donation (yrs); median (range) | 50.3 (29–65) | — | — |
| Time since donation (yrs); median (range) | 7 (4–14) | — | — |
| eGFR (ml/min/1.73m$^2$); median (range) | 60 (44–86) | — | — |
| School attendance (yrs); median (range) | 11 (8–13) | 12 (9–13) | .217 |
| Hypertension; N (%) | 23 (38.3) | 4 (10.0) | **.002** |
| Coronary heart disease; N (%) | 1 (1.7) | 1 (2.5) | .771 |
| Hyperlipidemia; N (%) | 9 (15.0) | 1 (2.5) | **.041** |
| Diabetes mellitus; N (%) | 1 (1.7) | 1 (2.5) | .771 |
| Hypothyroidism; N (%) | 7 (11.7) | 1 (2.5) | .098 |
| PCS, median (range) (50, SD 10) | 53.1 (17–62.3) | 54.1 (27.4–59.2)* | .067 |
| MCS, median (range) (50, SD 10) | 56.0 (26.6–64.3) | 55.0 (34–74)* | .375 |
| MFI, general score, median (range) (0–20) | 8.0 (4–16) | — | — |
| PHQ-9; median (range) (0–27) | 2.0 (0–14) | — | — |
| GAD-7; median (range) (0–21) | 1.5 (0–14) | — | — |
| SCL-9 (GSI); median (range) | 0.7 (0–2.4) | — | — |

GAD-7 = Generalized Anxiety Disorder Scale; MCS = Mental Component Scale (SF-12), MFI = Multidimensional Fatigue Inventory; PCS = Physical Component Scale (SF-12); PHQ-9 = Patient Health Questionnaire-Depression Scale; SCL-9 = Short form of the Symptom Checklist 90; yrs = years

[1]Student's t-test

*N = 39; bold print = significant result.

significantly more misses in the subtest working memory than controls, had slower reaction times (RT) in the subtests divided attention and incompatibility and were less successful in all results of the cancelling d test. The effect sizes ranged from small to large with a large effect size for the difference in misses in the "working memory" test ($d$ = .824). There was no difference between LKD and controls in most of the memory tests. However, donors exhibited worse results for nonsense figures than controls in the RFT (Table 2).

Regarding the number of abnormal test results, 8 (13.3%) of the donors and 3 (7.5%) of the controls had an attention test sum score above 0.4 which was considered as clinically relevant impaired attention; this difference was not statistically significant (Table 3). With regard to the individual tests, the comparison of the number of abnormal test results fits well with the comparison of the raw values between LKD and controls (Table 2). A higher percentage of LKD than of controls exhibited errors and misses in the working memory subtest and cancelling d test. In the subtest divided attention, donors responded more often abnormally slow towards auditory stimuli, but made less often mistakes. They also had more often a prolonged RT in the "valid" condition of the subtest covert shift of attention. In terms of memory function, donors and controls obtained comparable numbers of abnormal results in all subtests (Table 3).

## Associations between cognitive test results and other variables in LKD

Overall, donors' mean eGFR decreased from 96.2 (± 10.1) ml/min/1.73m$^2$ pre-donation to 56.8 (± 8.8) ml/min/1.73m$^2$ immediately post-donation, then increased to 59.9 (± 10.9) ml/min/1.73m$^2$ at first visit 2–6 weeks after donation and to 61.3 (± 9.6) ml/min/1.73m$^2$ at the time of assessment. At the time of testing 28 LKD had eGFR values < 60 (48.3%) and 30 (51.7%) of ≥ 60 ml/min/1.73m$^2$; the values of two LKD were missing.

**Table 2. Comparison of cognitive test results between living kidney donors (LKD) and controls.**

| | LKD N = 60 | | Controls N = 40 | | p-value | Effect size |
|---|---|---|---|---|---|---|
| | median | 25th; 75th percentile | median | 25th; 75th percentile | U- or t-test | Cohen's d |
| Attention test sum score | 0.156 | (0.063; 0.297) | 0.125 | (0.063; 0.188) | .461 | .148 |
| TAP Alertness | | | | | | |
| RT without warning sound (ms)* | 280.0 | (246.5; 311.3) | 267.5 | (232.8; 296.8) | .129 | .307 |
| RT with warning sound (ms)* | 259.0 | (241.3; 282.8) | 256.5 | (231.0; 282.4) | .229 | .242 |
| TAP Working Memory | | | | | | |
| RT (ms)* | N = 56; 587.5 | (480.3; 734.0) | 553.5 | (490.0; 623.0) | .471 | .148 |
| Errors (N)* | N = 57; 2.0 | (0.5; 5.0) | 1.0 | (0; 3.8) | .232 | .244 |
| Misses (N)* | N = 57; 3.0 | (1.0; 5.0) | 1.0 | (0; 2.0) | < .001 | .824 |
| TAP Crossmodal Integration | | | | | | |
| RT (ms)* | N = 59; 437.0 | (390.0; 520.0) | 432.5 | (398.9; 460.5) | .512 | .132 |
| Errors (N)* | N = 59; 0.0 | (0: 2.0) | 1.0 | (0; 1.0) | .948 | .013 |
| TAP Flexibility | | | | | | |
| RT (ms)* | N = 59; 787.0 | (640.0; 948.0) | 742.3 | (645.8; 855.0) | .322 | .200 |
| Errors (N)* | N = 59; 1.0 | (0; 3.0) | 1.50 | (0; 3.8) | .835 | .042 |
| TAP Divided Attention | | | | | | |
| RT auditive (ms)* | N = 58; 652.0 | (588.0; 720.0) | 623.3 | (543.5; 667.3) | **.026** | .464 |
| RT visual (ms)* | N = 58; 814.0 | (763.0; 907.0) | 848.5 | (798.8; 887.3) | .196 | .264 |
| Errors (N)* | N = 58; 1.0 | (0; 2.0) | 1.0 | (0; 3.0) | .153 | .292 |
| Misses (N)* | N = 58; 2.0 | (0; 3.0) | 2.0 | (1.0; 3.0) | .631 | .097 |
| TAP Go/No Go | | | | | | |
| RT (ms)* | N = 59; 454.0 | (402.0; 513.0) | 442.0 | (413.8; 471.8) | .564 | .116 |
| Errors (N)* | N = 59; 0.0 | (0; 1.0) | 0.0 | (0; 1.0) | .770 | .059 |
| TAP Incompatibility | | | | | | |
| RT (ms)* | N = 58; 524.5 | (486.5; 584.0) | 484.0 | (417.6; 545.4) | **.004** | .601 |
| Errors (N)* | N = 58; 1.0 | (0; 6.25) | 2.0 | (1.0;4.0) | .682 | .083 |
| TAP Covert Shift of Attention | | | | | | |
| RT, valid right (ms) | N = 58; 357.5 | (315.8; 410.0) | 339.5 | (300.5; 380.8) | .116 | .322 |
| RT, valid left (ms) | N = 58; 354.0 | (315.8; 407.3) | 340.0 | (312.5; 382.5) | .246 | .236 |
| RT, invalid right (ms) | N = 58; 409.5 | (369.0; 471.8) | 386.5 | (324.3; 458.0) | .133 | .307 |
| RT, invalid left (ms) | N = 58; 395.5 | (341.8; 459.0) | 377.0 | (329.3; 444.0) | .219 | .250 |
| d2 Test of Sustained Attention | | | | | | |
| Errors (%)* | 6.4 | (3.0; 9.5) | 3.7 | (2.1; 6.4) | **.016** | .498 |
| Error-corrected total number (N)* | 356.5 | (313.5; 402.3) | 395.0 | (354.0; 448.5) | **.007**[1] | .553 |
| Capacity of concentration (N) | 131.0 | (113.0;157.8) | 151.0 | (138.0;173.8) | **.003**[1] | .595 |
| Recurring Figures Memory Test | | | | | | |
| Nonsense (raw value) | 3.0 | (0; 7.0) | 6.0 | (2.0; 10.75) | **.022** | .472 |
| Geometric (raw value) | 17.0 | (15.0; 18.0) | 17.0 | (16.0; 19.0) | .103 | .058 |
| Word Figure Memory Test | | | | | | |
| Words (raw value) | N = 59; 11.0 | (7.0; 14.9) | 12.0 | (8.3; 15.0) | .490[1] | .142 |
| Figures (raw value) | N = 59; 13.0 | (8.0; 17.0) | 14.0 | (10.3; 18.0) | .303[1] | .212 |

Number of subjects considered for calculations differ because some donors were not able to complete all subtests. In these cases, reaction times and numbers of misses and errors were not available for calculation. ms = milliseconds; RT = reaction time; TAP = Test of Attentional Performance

Mann-Whitney-U tests except for

[1] = Student's t test

* = results are integrated in the attention test sum score; bold print = significant result.

**Table 3. Number and percentage of LKD and controls with abnormal test results.**

| | LKD N = 60 | | Controls N = 40 | | p-value | Effect size |
|---|---|---|---|---|---|---|
| | N | % | N | % | $X^2$ test | phi |
| Attention test sum score >0.4 | 8 | 13.3 | 3 | 7.5 | .361 | -.091 |
| TAP Alertness | | | | | | |
| RT without warning sound | 17 | 28.3 | 14 | 35.0 | .480 | .071 |
| RT with warning sound | 19 | 31.8 | 12 | 30.0 | .860 | -.018 |
| TAP Working Memory | | | | | | |
| RT | 8 | 13.3 | 2 | 5.0 | .174 | -.136 |
| Errors | 16 | 26.7 | 4 | 10.0 | **.041** | -.204 |
| Misses | 17 | 28.3 | 4 | 10.0 | **.027** | -.221 |
| TAP Crossmodal Integration | | | | | | |
| RT | 20 | 33.3 | 7 | 17.5 | .081 | -.175 |
| Errors | 10 | 16.7 | 3 | 7.5 | .182 | -.134 |
| TAP Flexibility | | | | | | |
| RT | 5 | 8.3 | 4 | 10.0 | .775 | .029 |
| Errors | 7 | 11.7 | 2 | 5.0 | .254 | -.114 |
| TAP Divided Attention | | | | | | |
| RT auditive | 29 | 48.3 | 9 | 22.5 | **.009** | -.261 |
| RT visual | 5 | 8.3 | 1 | 2.5 | .229 | -.120 |
| Errors | 8 | 13.3 | 12 | 30.0 | **.041** | .204 |
| Misses | 8 | 13.3 | 5 | 12.5 | .903 | -.012 |
| TAP Go/No Go | | | | | | |
| RT | 10 | 16.7 | 10 | 25.0 | .307 | .102 |
| Errors | 2 | 3.3 | 2 | 5.0 | .677 | .042 |
| TAP Incompatibility | | | | | | |
| RT | 8 | 13.3 | 5 | 12.5 | .903 | .102 |
| Errors (N) | 8 | 13.3 | 3 | 7.5 | .361 | -.091 |
| TAP Covert Shift of Attention | | | | | | |
| RT, valid right | 16 | 26.7 | 4 | 10.0 | **.041** | -.204 |
| RT, valid left | 14 | 23.3 | 6 | 15.0 | .307 | -.102 |
| RT, invalid right | 20 | 33.3 | 8 | 20.0 | .146 | -.102 |
| RT, invalid left | 15 | 25.0 | 7 | 17.5 | .375 | -.089 |
| d2 Test of Sustained Attention | | | | | | |
| Errors (%) | 3 | 5.0 | 1 | 2.5 | .532 | -.063 |
| Error-corrected total number | 17 | 28.3 | 4 | 10.0 | **.027** | -.221 |
| Capacity of concentration | 11 | 18.3 | 2 | 5.0 | **.052** | -.194 |
| Recurring Figures Memory Test | | | | | | |
| Nonsense | 7 | 11.7 | 1 | 2.5 | .098 | -.166 |
| Geometric | 1 | 1.7 | 1 | 2.5 | .771 | .029 |
| Word Figure Memory Test | | | | | | |
| Words | 5 | 8.3 | 3 | 7.5 | .880 | -.015 |
| Figures | 9 | 15.0 | 5 | 12.5 | .724 | -.035 |

RT = reaction time; TAP = Test of Attentional Performance; bold print = significant result.

In univariate linear regression analyses there were no associations between the attention test sum score and sex, age, time since donation, kidney function (eGFR: β = 0.07; 95% CI = -.004 to .007, p = .59 and delta eGFR pre- to post-donation: β = -0.08; 95% CI = -0.01 to 0.01, p = .54), fatigue, quality of life and levels of distress, depression and anxiety, or any concomitant

disease. Years of school attendance was the only significantly associated factor for the attention test sum score (β = -0.43; 95% CI = -0.09 to -0.03, p = .001). The longer LKD had attended school, the lower (more normal) were their attention test sum scores. Years of education alone explained 17.5% of the variance of the attention test sum score (corrected $R^2$). Since we did not find significant associations other than educational level we did not perform multivariate regression models.

## Discussion

To our knowledge this is the first study assessing cognitive functioning in German LKD several years after donation. Our hypothesis was that LKD would show impaired cognitive functioning compared to norm data and to healthy controls which would correlate with kidney function and mental status. Sixty donors who had undergone kidney donation at Hannover Medical School (MHH) between 2003 and 2012 and were representative for the respective cohort of donors at MHH participated in the study.

There were no differences between LKD and controls in most of the individual results of the cognitive tests applied and specifically in the composite attention test sum score. Eight out of 60 donors as compared to three out of 40 controls achieved an abnormal attention test sum score of >.0.4 which is considered to indicate a clinically relevant cognitive impairment. However, the level of performance of LKD was below normal (compared to norms) and significantly different from controls in some results of the TAP tests working memory, divided attention, covert shift of attention, incompatibility, and in the cancelling d test. This might suggest some low-grade cognitive impairment, but considering the large number of tests and the lack of difference in the attentional composite score our data do not support the presence of severe cognitive impairment in LKD compared to healthy controls. Regarding the pattern of abnormal cognitive test results, we found similarities and differences between our sample of LKD and individuals with CKD in general population samples. General and visual attention and concentration were impaired in the LKD in our study similarly to individuals with CKD in population samples. However, in contrast to the LKD in our study, individuals with CKD frequently also exhibit memory impairment [10–12, 34].

With regard to the association between eGFR and cognitive functioning, cross-sectional population studies have reported conflicting results. Hailpern et al. [33] showed that moderate CKD (eGFR of 30–59 ml/min/1.73m$^2$) was associated with poorer results in tests on visual attention and learning and concentration. In another study, subjects with a eGFR <60 ml/min/1.73m$^2$ performed worse in tests of visual and spatial organization and memory as well as scanning and tracking compared to subjects with a eGFR ≥60 ml/min/1.73m$^2$ [11, 12], while working memory and verbal episodic memory were not impaired. In contrast, Davey et al. [34] showed in a long-term investigation, that with a clinically significant decline in eGFR (>3ml/year) subjects performed worse in the global composite of cognitive performances and in similarities and verbal episodic memory. In the Brain in Kidney Disease Study [10] participants with a eGFR between 30 and 59 ml/min/1.73m$^2$ performed 0.2 to 0.5 standard deviations below norms in all tested domains of attention and memory. However, consistent with the results of the present study, others did not observe associations between performance on cognitive tests and eGFR or measured GFR in general population samples [35–37]. In the general population, low GFR may be a sign of systemic atherosclerosis. However, LKD develop reduced kidney function through a different mechanism, and their clinical and prognostic significance remains uncertain [2].

LKD with abnormal attention test sum score differed in terms of educational level from LKD with normal scores although norms adjusted for education were used for scoring if

applicable. Former studies have shown that a higher educational level might have a protective effect on cognitive function in patients with CKD [11]. This concept has been pursued in patients with metabolic disorders such as CKD or liver cirrhosis, but has also been discussed for example for Alzheimer's dementia where intelligence and education have been considered supporting cognitive reserve [38]. Further research will be necessary to confirm the positive effect of education on attention and memory in LKD.

We hypothesised that an impaired cognitive function was correlated to higher fatigue scores, lower quality of life, as well as higher levels of distress, depression and anxiety. Depression is known to affect cognitive functioning [39, 40]. In our study; however, there was no difference regarding symptoms of fatigue, distress, depression, and anxiety between the patients with and without cognitive impairment. In our study we excluded participants with a diagnosed mental disorder; thus, in both groups the mean scores for the scales were clearly within population norms, indicating low symptom severity which might explain the lack of association with cognitive functioning.

## Strengths and limitations of the study

This is the first pilot study investigating cognitive functioning in LKD using a comprehensive state-of-the-art cognitive test battery for the assessment of attention and memory and using a methodology that has been shown to be reliable in other patient populations [21–23]. The test results were compared to norms adjusted for age, sex and education and we also included a healthy control group.

However, there are several limitations. Firstly, due to the cross-sectional design of our pilot study without data on cognitive functions before donation, we do not know if cognitive functioning changed after nephrectomy in our LKD. There is evidence that LKD are usually physically and mentally healthier compared to the general population before donation [7, 41]. This might also be true for neurocognitive functioning. Ultimately, only prospective studies will allow to determine whether LKD are confronted with post-operative cognitive decline.

The groups were fairly small and there could have been a selection bias, because we tested a convenience sample of 60 LKD out of 184 participants who met inclusion criteria for this substudy. Also, the healthy control group was not matched. Because LKD are a selected sample of especially healthy individuals, the comparison with normative samples has been criticized [41].

## Conclusions

In summary, 4 to 14 years after the kidney donation, we found some evidence that cognitive functioning might show some impairment in LKD compared to healthy controls even though a composite attention test sum score did not show significant differences. Test results were not associated with age, sex, kidney function, time since transplantation, quality of life, and mental status. These findings need further evaluation especially in longitudinal studies that address the intra-individual change of cognitive function from before to after donation. Also, a comparison of LKD and patients with modest early-stage CKD may help understanding the mechanism how kidney function affects attention and memory functions.

## Supporting information

**S1 File.**
(DOCX)

**S2 File.**
(DOCX)

**S1 Data.**
(CSV)

**S2 Data.**
(CSV)

## Acknowledgments

This paper is dedicated to Faikah Güler, who died on March 20, 2020. Faikah Güler was dedicated to her clinical and scientific work, but especially to her patients. We will keep her in fond memory.

We also thank Karl-Heinz Heiringhoff for excellent IT support and Hanna Prominski for help with management of the study.

## Author Contributions

**Conceptualization:** Faikah Gueler, Martina de Zwaan, Karin Weissenborn.

**Data curation:** Marie Mikuteit.

**Formal analysis:** Marie Mikuteit, Karin Weissenborn.

**Funding acquisition:** Marie Mikuteit, Karin Weissenborn.

**Investigation:** Marie Mikuteit, Iris Pollmann, Henning Pflugrad, Meike Dirks.

**Methodology:** Faikah Gueler, Martina de Zwaan.

**Project administration:** Martina de Zwaan, Karin Weissenborn.

**Supervision:** Faikah Gueler, Martina de Zwaan, Karin Weissenborn.

**Writing – original draft:** Marie Mikuteit.

**Writing – review & editing:** Faikah Gueler, Iris Pollmann, Henning Pflugrad, Meike Dirks, Martina de Zwaan, Karin Weissenborn.

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
