## [Decision Letter · Decision Letter 0]

21 Dec 2021

PONE-D-21-31779Assessment of cognitive functioning after living kidney donation: a cross-sectional pilot study.PLOS ONE

Dear Dr. de Zwaan,

Thank you for submitting your manuscript to PLOS ONE. After careful consideration, we feel that it has merit but does not fully meet PLOS ONE’s publication criteria as it currently stands. Therefore, we invite you to submit a revised version of the manuscript that addresses the points raised during the review process.

ACADEMIC EDITOR: Based on the review by two experts in the field, I would recommend to thoroughly revise the MS based on the comments, which are minor, but need to be reassessed by the reviewer. Please provide point-by-point comments to the reviewers comments and questions. This is not a guarantee for acceptance of the revised MS.

We look forward to receiving your revised manuscript.

Kind regards,

Frank JMF Dor, M.D., Ph.D., FEBS, FRCS

Academic Editor

PLOS ONE

Journal Requirements:

Reviewers' comments:

Reviewer's Responses to Questions

**Comments to the Author**

1. Is the manuscript technically sound, and do the data support the conclusions?

Reviewer #1: Yes

Reviewer #2: Yes

2. Has the statistical analysis been performed appropriately and rigorously? 

Reviewer #1: Yes

Reviewer #2: Yes

3. Have the authors made all data underlying the findings in their manuscript fully available?

Reviewer #1: Yes

Reviewer #2: Yes

4. Is the manuscript presented in an intelligible fashion and written in standard English?

Reviewer #1: Yes

Reviewer #2: Yes

5. Review Comments to the Author

Reviewer #1: Thank you for asking me to review this manuscript looking at cognitive function in living kidney donors.

I have a few points which I feel require further clarification:

1. Was there a reason why the quality of life measures were not given to the control group?

2. Did you do a sample size calculation?

3. Why were patients with mental health problems excluded?

I have minor comments re the language:

Line 50: the term ‘deceased donation’ is considered more appropriate than post-mortem donation

Line 51: I wouldn’t call rehospitalisation a side-effect – it is a complication

Line 53: Long run ought to be long term

Line 56-59: I am unsure of what is meant by “A meta-analysis could show…”

Line 154: There is a typo – der instead of the

Line 244: I would remove the word ‘group’

Line 253-258: These three sentences are written poorly and are confusing in what they are trying to say. Please rephrase these.

Reviewer #2: This is a thorough and diligently performed study, well designed, with a clear research questions, clearly and accurately written. I have no comments for improvement. The study can be published as it is. If I would have a minor comment it would refer to how the authors explain the reduced scores in working memory, slower reaction times and divided attention, meaning how do they explain the relationship between eGFR and these low scores... Do they assume it is a physical parameter making the difference? Or could it be that living kidney donors have an additional "worry" as in an affective rumination going on permanently related to the loss of their kidney and a potential risk for kidney function due to donation that affects working memory? Please consider this maybe unorthodox idea more of a comment in terms of an informal academic exchange rather than a comment for revision or improvement.

6. PLOS authors have the option to publish the peer review history of their article (what does this mean?). If published, this will include your full peer review and any attached files.

Reviewer #1: No

Reviewer #2: No

---

## [Author Response · Author response to Decision Letter 0]

4 Jan 2022

Editor

With regard to the data sharing requirements of PlosOne we now provide all relevant data as Supporting Information files.

Reviewer #1

I have a few points, which I feel require further clarification:

1. Was there a reason why the quality of life measures were not given to the control group?

The main quality of life instrument, the SF-12 (with the subscales PCS and MCS), was actually completed also by the controls (see Table 1). We found no difference between groups and both groups showed scores within the normal range for quality of life. Only the living donors filled out the other self-rating measures (fatigue, depressive and anxiety symptoms) as part of a more comprehensive assessment battery within the follow-up study.

2. Did you do a sample size calculation?

This investigation was designed as a pilot study. Since this is the first study investigating cognitive functioning in individuals after living kidney donation, we did not have a solid database for a power calculation.

3. Why were patients with mental health problems excluded?

Mental disorders such as depression can affect cognitive performance. We wanted to eliminate one strong potentially confounding factor. As mentioned in the discussion, “Depression is known to affect cognitive functioning”. We supported this statement by citing two references:

39. Ahern E, Semkovska M. Cognitive functioning in the first-episode of major depressive disorder: a systematic review and meta-analysis. Neuropsychology. 2017;31:52–72.

40. Liu J, Liu B, Wang M, Ju Y, Dong Q, Lu X, et al. Evidence for progressive cognitive deficits in patients with major depressive disorder. Front Psychiatry. 2021;12:627695.

I have minor comments re the language:

We thank the reviewer for carefully reading and editing the manuscript. We followed the suggested changes.

Line 50: the term ‘deceased donation’ is considered more appropriate than post-mortem donation.

We agree with the reviewer’s suggestion and substituted “post-mortem” with “deceased”.

Line 51: I wouldn’t call rehospitalisation a side-effect – it is a complication

We changed the wording accordingly.

Line 53: Long run ought to be long term

We changed “long run” to “long term”.

Line 56-59: I am unsure of what is meant by “A meta-analysis could show…”

We re-phrased the sentence: “A systematic review revealed that most, but not all, cross-sectional and longitudinal studies suggest an association between cognitive impairment and CKD.”

Line 154: There is a typo – der instead of the

We corrected the typo. Thanks for catching that.

Line 244: I would remove the word ‘group’

We omitted the word “group”

Line 253-258: These three sentences are written poorly and are confusing in what they are trying to say. Please rephrase these.

The sentences were re-written and readability should be improved.

“Regarding the pattern of abnormal cognitive test results, we found similarities and differences between our sample of LKD and individuals with CKD in general population samples. General and visual attention and concentration was impaired in the LKD in our study similarly to individuals with CKD in population samples. However, in contrast to the LKD in our study, individuals with CKD frequently also exhibit memory impairment [10-12, 34].”

Reviewer #2

This is a thorough and diligently performed study, well designed, with a clear research questions, clearly and accurately written. I have no comments for improvement. The study can be published as it is. If I would have a minor comment, it would refer to how the authors explain the reduced scores in working memory, slower reaction times and divided attention, meaning how do they explain the relationship between eGFR and these low scores... Do they assume it is a physical parameter making the difference? Or could it be that living kidney donors have an additional "worry" as in an affective rumination going on permanently related to the loss of their kidney and a potential risk for kidney function due to donation that affects working memory? Please consider this maybe unorthodox idea more of a comment in terms of an informal academic exchange rather than a comment for revision or improvement.

We thank the reviewer for his/her supporting comments. It might be possible that “worries” influence cognitive performance; however, in our experience with working with living donors this is not a frequent problem. In addition, as can be seen in Table 1, quality of life, depression, anxiety, fatigue and levels of distress were within normal ranges.

---

## [Decision Letter · Decision Letter 1]

8 Feb 2022

Assessment of cognitive functioning after living kidney donation: a cross-sectional pilot study.

PONE-D-21-31779R1

Dear Dr. de Zwaan,

We’re pleased to inform you that your manuscript has been judged scientifically suitable for publication and will be formally accepted for publication once it meets all outstanding technical requirements.

Kind regards,

Frank JMF Dor, M.D., Ph.D., FEBS, FRCS

Academic Editor

PLOS ONE

Additional Editor Comments (optional):

Thank you for addressing the comments.

Reviewers' comments:

Reviewer's Responses to Questions

**Comments to the Author**

1. If the authors have adequately addressed your comments raised in a previous round of review and you feel that this manuscript is now acceptable for publication, you may indicate that here to bypass the “Comments to the Author” section, enter your conflict of interest statement in the “Confidential to Editor” section, and submit your "Accept" recommendation.

Reviewer #1: All comments have been addressed

Reviewer #2: All comments have been addressed

2. Is the manuscript technically sound, and do the data support the conclusions?

Reviewer #1: Yes

Reviewer #2: Yes

3. Has the statistical analysis been performed appropriately and rigorously? 

Reviewer #1: Yes

Reviewer #2: Yes

4. Have the authors made all data underlying the findings in their manuscript fully available?

Reviewer #1: Yes

Reviewer #2: Yes

5. Is the manuscript presented in an intelligible fashion and written in standard English?

Reviewer #1: Yes

Reviewer #2: Yes

6. Review Comments to the Author

Reviewer #1: Thank you for addressing my comments. I have nothing further to add.

Reviewer #2: The authors have addressed the minor comments raised by the reviewers. No other comments from my side. The paper can be accepted.

7. PLOS authors have the option to publish the peer review history of their article (what does this mean?). If published, this will include your full peer review and any attached files.

Reviewer #1: No

Reviewer #2: No

---

## [Editor Report · Acceptance letter]

11 Feb 2022

PONE-D-21-31779R1 

Assessment of cognitive functioning after living kidney donation: a cross-sectional pilot study. 

Dear Dr. de Zwaan:

I'm pleased to inform you that your manuscript has been deemed suitable for publication in PLOS ONE. Congratulations! Your manuscript is now with our production department. 

Kind regards, 

on behalf of

Dr. Frank JMF Dor 

Academic Editor

PLOS ONE